# Immune Checkpoint Inhibitors in “Special” NSCLC Populations: A Viable Approach?

**DOI:** 10.3390/ijms241612622

**Published:** 2023-08-09

**Authors:** Giuseppe Bronte, Donato Michele Cosi, Chiara Magri, Antonio Frassoldati, Lucio Crinò, Luana Calabrò

**Affiliations:** 1Department of Clinical and Molecular Sciences (DISCLIMO), Università Politecnica Delle Marche, Via Tronto 10/A, 60121 Ancona, Italy; 2Clinic of Laboratory and Precision Medicine, National Institute of Health and Sciences on Ageing (IRCCS INRCA), 60124 Ancona, Italy; 3Department of Oncology, University Hospital of Ferrara, 44124 Cona, Italy; 4Department of Medical Oncology, IRCCS Istituto Romagnolo Per Lo Studio Dei Tumori (IRST) “Dino Amadori”, 47014 Meldola, Italy; 5Department of Translational Medicine, University of Ferrara, 44121 Ferrara, Italy

**Keywords:** NSCLC, immunotherapy, immune checkpoint inhibitors, special populations

## Abstract

Over the last decade, the therapeutic scenario for advanced non-small-cell lung cancer (NSCLC) has undergone a major paradigm shift. Immune checkpoint inhibitors (ICIs) have shown a meaningful clinical and survival improvement in different settings of the disease. However, the real benefit of this therapeutic approach remains controversial in selected NSCLC subsets, such as those of the elderly with active brain metastases or oncogene-addicted mutations. This is mainly due to the exclusion or underrepresentation of these patient subpopulations in most pivotal phase III studies; this precludes the generalization of ICI efficacy in this context. Moreover, no predictive biomarkers of ICI response exist that can help with patient selection for this therapeutic approach. Here, we critically summarize the current state of ICI efficacy in the most common “special” NSCLC subpopulations.

## 1. Introduction

In the last 15 years, one of the major advances in clinical oncology has been the introduction of immunotherapy (IO) to the treatment of a significant number of cancers, particularly lung cancer.

Since 2013, several randomized trials have shown the superiority of programmed cell death protein (PD)-1/PD-ligand (L)1 immune checkpoint inhibitors (ICIs) in terms of overall survival (OS), in both the first- and second-line treatments of metastatic non-small-cell lung cancer (NSCLC); in addition, several anti-PD1-PD-L1 drugs have become integrant or exclusive components of the treatment strategy for unresectable or metastatic advanced NSCLC [1,2,3,4].

However, in all randomized trials of ICIs ± chemotherapy vs. chemotherapy alone, in both first- and second-line treatments, the real clinical benefit in terms of OS at 5 years of follow up has resulted in being limited to a restricted number of patients; this has ranged between 16 and 20% of metastatic stage IV NSCLC patients [5,6]. To date, we do not have consistent reliable biomarkers to identify these patients beyond PD-L1 expression or mutational tumor burden (TMB), both of which present significant insufficiency and heterogeneity. For this reason, IO, in the absence of specific selective biomarkers, has been registered as the preferred treatment for several cancers of different histology, including melanoma, kidney cancer, lung cancer, gastrointestinal cancer, bladder cancer, head and neck cancer, and hepatocellular cancer [7]. Despite this very broad label indication, IO could not be suitable for all NSCLC patients; this is particularly true with NSCLC.

In several randomized clinical trials, IO has shown a significant clinical benefit in comparison to platinum-based chemotherapy in patients with a metastatic NSCLC and PD-L1 expression over 50%; it has also been shown that there is a significant linear correlation in clinical outcome between the PD-L1 tumor proportion score (TPS) and the magnitude of benefit.

The combination of IO and chemotherapy in first-line treatment has been approved as the potential treatment of choice, but in a recent FDA pooled analysis of 12 randomized clinical trials (RCTs) on patients with a PD-L1 of ≥50% treated with FDA-approved chemo-IO regimens, it was shown that the majority of subgroups may have had improved outcomes compared to those on IO-only regimens. However, patients ≥ 75 years of age may not have better survival outcomes with chemo-IO in comparison to IO only [8]; in the same pooled analysis, IO alone, IO in combination with chemotherapy, or chemotherapy alone provided similar results regardless of the KRAS mutation status [9].

Traditionally, elderly patients, those with a poor performance status, and those with brain metastases are considered to be ‘special’ populations of NSCLC patients, because of their poor prognosis and the possibility that chemotherapy’s efficacy could be limited. The purpose of this paper is to overview if these categories can also differentially impact on IO’s efficacy. We also have to add another special population, such as never smokers and/or oncogene-addicted patients, because an addiction to specific oncogenes could limit the chance of producing neoantigens that could be targeted by the immune system. Moreover, we discuss the biomarkers for immune response to try to understand whether potential predictive factors can select immune-sensitive patients within those special populations.

## 2. IO in Never-Smoker and/or Oncogene-Addicted NSCLC Patients

Since the discovery of the epidermal growth factor receptor (EGFR) by Stanley Cohen (Nobel prize, 1986), a number of studies and clinical trials have developed the paradigm of precision medicine based on targetable genomic alterations. Oncogenic DNA mutations and DNA tumor rearrangements are the basis for molecular-driven therapy in lung adenocarcinoma. Since 2004, oncogene addiction has been demonstrated in terms of DNA mutations in the EGFR (exon 19–21); B-RAF (V600E); MET (exon 14 skip mutations); intragenic insertions in EGFR and ERB-B2 (exon 20); or fusion rearrangements in ALK, ROS1, RET, NTRK, and NRG1 in patients with lung adenocarcinoma. Proto-oncogenes induce cell proliferation and apoptosis inhibition through the activation of different specific tyrosine kinases (TKs). Therefore, a number of TKI inhibitors have been developed and approved as the preferred treatments for oncogene-addicted lung adenocarcinomas [10].

TKIs consist of oral small molecules and have shown superiority over chemotherapy in many randomized clinical trials; consequently, they represent the first line of treatment in EGFR, ALK, ROS1, and MET proto-oncogene-driven tumors, with a response rate ranging between 60 and 80% and a clear superiority in PFS and OS [11,12]. However, following years of critical evaluation of the results of TKI-based targeted therapies in all specific subsets of lung adenocarcinomas, some concerns have been raised: despite their high activity, the complete response rate is usually less than 5%, and almost all patients soon or later develop a resistance to TKIs and experience progressive disease. A large intratumor heterogeneity for somatic mutations, rearrangements, and gene amplifications was observed in a prospective cohort study on the whole-exome sequencing of 100 early stages of resected NSCLC [13]. This genomic heterogeneity and chromosomal instability can explain the different clinical outcomes and variable phenotypes that are present following the same treatment and in the same oncogene addiction. Currently, platinum-based chemotherapy is the only consistent second-line treatment; however, it is generally inadequate for patients with oncogene-addicted metastatic NSCLC. Immunotherapy seems to work poorly on oncogene-addicted tumors. In all the RCTs comparing IO and chemotherapy as both first- and second-line treatments, a forest plot analysis showed that oncogene addiction, mainly that of EGFR mutations and ALK rearrangements, and a never-smoker status represented the variables that were more consistently associated with a chemotherapy benefit than IO [14,15]. The poor activity of IO in these situations can be explained by the absence of smoking exposure in all these cases, considering that a never-smoker status accounts for most oncogene-addicted tumors. NSCLC in never smokers has long been considered a “different disease” [16] with a lower median age, a slight prevalence in females, widespread clinical dissemination despite a good performance status (PS), and the increasing presence of oncogene addiction. In never smokers and oncogene-addicted tumors, the number of somatic TMBs is lower, and the PD-L1 upregulation, which is often present, has a different biological role [17]. The tumor microenvironment is mostly poorly inflamed and results in a cold immune phenotype, which can explain the modest activity of IO [18].

ICI treatment in oncogene-driven tumors has consistently yielded poor clinical results, and when combined with TKIs in EGFR, ALK, and ROS1 patients, it has been shown to have highly relevant toxicities, such as intestinal pneumonitis, liver failure, cutaneous dermatitis up to Stevens–Johnson syndrome, and systemic fever [19].

In the IMMUNOTARGET registry, in which data from 24 international institutions are collected, the overall response rate (ORR) to anti-PD-1 agents in 115 EGFR-mutated patients was 12%, and the median progression-free survival (mPFS) was only 2.1 months; in the same registry, in 19 ALK-rearranged patients, no activity at all was reported with ICI monotherapy after TKI failure, with an mPFS of 2.5 months [16]. In the same study, the patient population treated with IO, who harbored MET exon 14 skipping mutations, RET rearrangements, or BRAF mutations, showed modest activity, with ORRs of 16%, 6%, and 24%, and poor mPFSs of 3.4, 2.1, and 3.1 months, respectively [20].

Furthermore, in a meta-analysis of randomized trials comparing different PD-L1 inhibitors to docetaxel as the control arm in second-line treatment after the failure of platinum-based chemotherapy, never-smoker patients with an activating EGFR mutation did not benefit from immunotherapy (HR: 1.05; 95% CI 0.70–1.55) [21].

The poor activity of immunotherapy in EGFR-mutant patients was confirmed in a prospective phase II study, in which pembrolizumab was used as the first-line treatment in EGFR-mutant patients with a PD-L1 TPS of >10%. The trial was prematurely closed after enrolling 11 patients because of the lack of activity and concern about safety, which arose with the development of interstitial lung disease (ILD) and two reported deaths in the first 6 months of treatment [22].

In a recent retrospective study performed at the Memorial Sloan Kettering Cancer Center and the Dana–Farber Cancer Institute, in 147 patients with MET exon 14 skipping alterations of any stage, the response to pembrolizumab in 24 patients was poor (ORR: 17%) and the PFS was a very short 1.9 months, independent of PD-L1 expression and with a lower TMB in comparison to non-addicted NSCLC [23].

The idea of combining TKIs and IO to improve the outcome of oncogene-addicted patients has brought about unexpected safety problems in both the concurrent and sequential administration of ICIs and TKIs, with different toxicity profiles in EGFR-mutated and ALK-rearranged patients. Indeed, ILD, pneumonitis, liver toxicities, cutaneous erythema, and dermatitis have been reported at an increased rate in different trials of combination treatments with heterogeneous behavior, and most trials have been stopped due to toxicities and a lack of activity [24,25].

In daily practice, in addition to clinical trials, the sequential administration of IO and TKIs can occur on the basis of the overexpression of PD-L1 and delayed information regarding NGS mutational status. In this case, the prolonged half-lives of PD-1/PD-L1 inhibitors can induce enhanced toxicity, which can manifest at the beginning of TKI-based treatment [26]. We have reported an exon 19 EGFR-mutated young female, who developed a systemic continuous fever, intestinal lung pneumonitis, and severe liver failure with Stevens–Johnson syndrome after the sequential administration of ICIs and TKIs. She fully recovered after one month of supportive treatment [27].

In the absence of molecular information, a wise recommendation is to not start immunotherapy in never smokers or former infrequent smokers with a high TPS. A lack of activity and unexpected enhanced toxicity has similarly been shown when combining ICIs and anti-ALK TKIs, mainly with a liver dose-limiting toxicity for both current and sequential administration [28].

According to these data, immunotherapy seems to have no role in the treatment of oncogene-addicted patients. Recently, however, in EGFR-mutated patients who failed first-line TKIs, a randomized trial has shown a significant PFS and OS benefit with the second-line combination of atezolizumab and carboplatin paclitaxel bevacizumab (PCB) vs. PCB alone (HR: 0.61 in PFS; HR: 0.31 in OS). This finding suggests an important potential interaction between immunotherapy and anti-angiogenesis [29], which may represent an important treatment achievement, with the caveat that it is the result of a post hoc subgroup analysis. For these reasons, IO is not currently taken into account as a treatment option in the first-line setting, in combination with TKI, or in a subsequent-line setting.

## 3. IO in Elderly NSCLC Patients

Owing to the aging population and progress in cancer treatment, the aged population with advanced NSCLC is increasing globally [30]. Conventionally, the elderly population includes those aged ≥ 70 years; around half of all patients with advanced NSCLC are within this population. By using a cutoff of 75 years, these patients represent around one third of the overall NSCLC population.

In recent years, the concept of being elderly has changed from a purely chronological evaluation to a more complex one, which also considers the biological age and functional and social status of the subjects. In this regard, several scales have been developed that can assess specific aspects, including function, comorbidities, quality of life, cognition, and emotional state [31], and a comprehensive geriatric assessment on decision making and treatment allocation is usually included in the multidisciplinary evaluation of elderly NSCLC patients.

Elderly patients can benefit from TKIs if they harbor oncogene driver mutations. In regard to EGFR-mutated patients, first-generation EGFR TKIs are favored over second-generation ones because of their lower grade 3–4 toxicity [32]. However, osimertinib is the best first-line option in older patients due to its good efficacy and tolerability [33,34]. Conversely, few ALK-rearranged elderly patients are represented in clinical trials. However, the ALEX study showed that, in elderly patients, alectinib achieved a better PFS and tolerability than crizotinib. More data are available for second-line ALK inhibitors in elderly patients, and age should not induce the exclusion of patients from receiving these kinds of drugs [33,35].

The introduction of ICIs to NSCLC patients has led to the reconsideration of the treatment paradigm in the elderly subpopulation, who, before the immunotherapy era, were often only candidates for best supportive care due to their ineligibility for chemotherapy.

However, though ICIs have demonstrated a better safety profile compared to chemotherapy, major concerns have been raised about their efficacy in elderly subjects. Indeed, immune senescence has been proposed as a process that favors cancer occurrence in older patients, but it can also limit the efficacy and impair the safety of anticancer treatments [29]. Nowadays, we know that aging may foster many mechanisms that affect the immune system. These include a reduction in bone marrow functions; a decrease in the size of other organs, such as the thymus, lymph nodes, and spleen; a reduction in the antigenic diversity of immune cells; a decrease in the co-stimulatory molecule expression on T lymphocytes; and a reduction in the antibody production of B lymphocytes. Moreover, “inflammaging”, a low-grade chronic inflammation state, is frequent with advanced age, and it is related to an increase in pro-inflammatory cytokines [36,37]. Conversely, Erbe and colleagues found biomarkers within a multi-omics database that were associated with an ICI response (such as TMB, ICI-related gene expression in selected tumors, and a more immune-stimulatory signaling TME) and were particularly enriched in tumors from older patients compared to those of their younger counterparts [38]. Moreover, the assumption that ICI therapy has a potentially reduced efficacy in the elderly has gradually faded away in light of the results generated from second-line, randomized clinical studies. Indeed, single-agent anti-PD-1 (nivolumab and pembrolizumab) and anti-PD-L1 (atezolizumab) have demonstrated a prolonged OS and long-term survival compared to docetaxel in platinum-refractory patients, including the subset of older patients aged over 65 years. Based on these results, ICI treatment has also been firmly established as a further line of treatment in the elderly population. However, in these studies, the OS was not improved in the restricted cohort of older patients aged over 75 years; these negative results could possibly have been affected by the small number of patients.

To further support the efficacy of ICIs in elderly patients, a meta-analysis of 17 randomized controlled trials that compared ICIs (nivolumab, pembrolizumab, or atezolizumab) to standard therapy (chemotherapy or targeted therapy) was conducted. The authors found comparable survival outcomes between younger patients and those older than 65 years [39]. A subsequent pooled analysis, including some of those trials, confirmed the same results with regard to the subgroup of patients aged > 75 years. The rates of severe toxicity in this pooled analysis were not different in this age subgroup, but its small number of patients could still have influenced this finding. We should presume that these very old patients were carefully selected according to their performance status; thus, they do not really represent the real-life population of NSCLC patients older than 75 years [40]. Another pooled analysis of studies comparing pembrolizumab and docetaxel in subsequent-line therapy achieved a survival benefit from pembrolizumab in both elderly and younger patients across each individual study [14,41,42]. Nevertheless, in a retrospective Japanese study, the enrolled 131 elderly NSCLC patients (aged ≥ 75 years) receiving a subsequent line of ICI monotherapy achieved an efficacy and safety level similar to that usually observed in younger patients [43].

Soon after, the use of ICIs rapidly moved to the frontline setting of NSCLC treatment. As for the second line, a restriction for age was not planned in trials on first-line ICIs versus chemotherapy, which included the study of pembrolizumab monotherapy for patients with a PD-L1 expression of ≥50% (Keynote-024); studies of pembrolizumab plus platinum-based chemotherapy (Keynote-189, Keynote-407); and the study of atezolizumab plus platinum-based chemotherapy and bevacizumab (IMPower150) [1,2,44,45]. In these trials, the authors described no differences in terms of overall survival when using a cutoff of 65 years of age. Similarly, patients aged ≥ 75 years did not show an improved survival with first-line chemo–IO over IO only [8]. Another trial (EMPOWER-Lung 1) in the first-line setting achieved a survival improvement for both younger and elderly patients (cutoff: 65 years) with cemiplimab monotherapy over platinum-based chemotherapy [46].

A real-life study included a larger group of NSCLC patients older than 70 years (*n* = 110), but the number of patients older than 80 years was still small (*n* = 16). Age did not significantly influence either the survival or toxicity in the patients receiving first- or second-line ICI-based therapy. The multivariate survival analysis of this study highlighted the performance status and number of metastatic sites as independent prognostic factors [47].

In a study including three cohorts of NSCLC patients treated with immunotherapy (*n* = 665), a high TMB made it possible to predict the durable clinical benefit only in patients aged less than 65 years [48]. This finding suggests that immune senescence processes could limit the neoantigen immunogenicity associated with a high TMB.

In a retrospective study comparing the uptake of systemic therapy before and after the availability of TKIs and IO, a small proportion of elderly patients received systemic therapy, but those treated achieved comparable survival outcomes to their younger counterparts [49]. Another retrospective study, evaluating patients who received anti-PD-1/PD-L1 in various age subgroups (<60, 60–69, 70–79, and ≥80 years), obtained differences in survival outcomes, but similar toxicity rates [50].

An international retrospective study of almost one thousand elderly patients (aged ≥ 80 years) also explored the tolerability of IO. They found that ICI-based treatment is effective in this population and mainly well tolerated, though the frequency of discontinuation owing to immune-related adverse effects increases with age [51]. Similarly, a meta-analysis of seven studies comparing IO and chemotherapy age subgroups (<65 and ≥65 years) achieved a similar overall survival benefit to anti-PD-1/PD-L1, but this result was not observed for the subgroup of patients aged > 75 years [52]. A similar finding was obtained in the meta-analysis by Sun et al. [53].

Overall, though efficacy was mainly confirmed across the data, some concerns still remain with regard to IO–chemotherapy combinations, especially in octogenarians. In this regard, a panel of experts have agreed on the need to expand clinical research with robust, real-world studies on elderly NSCLC patients. Currently, there is an ongoing phase III trial comparing atezolizumab plus carboplatin–paclitaxel vs. carboplatin–paclitaxel specifically in elderly patients (aged 70–89 years) with advanced NSCLC (ClinicalTrial ID: NCT03977194).

## 4. IO in Poor Performance Status (PS) Patients

The Eastern Cooperative Oncology Group (ECOG) PS is a scale used to estimate the impact of a malignancy on a patient’s daily living abilities, and it helps in the determination of the appropriate treatment and prognosis. It has scores ranging from 0 to 5, where 0 is the best and 5 means that the patient has died.

A poor PS may be induced by tumor burden and/or comorbidities. It is commonly believed that cancer patients with a tumor-related poor PS could benefit from treatments that induce a rapid tumor response. If a poor PS is mainly caused by comorbidities, therapy cannot change it.

Around one third of patients diagnosed with NSCLC have a poor PS (PS: 2–4) [54]. For those patients with metastatic NSCLC and a PS of 3–4 simultaneously, best supportive care is deemed to be the standard of care, because of the unfavorable risk–benefit ratio of chemotherapy and their short life expectancy (2–4 months from diagnosis).

Before the advent of targeted therapy and immunotherapy, chemotherapy was allowed for PS 2 patients when less toxic regimens were used [55,56]. The observation of the fast improvement in the PS of patients treated with TKIs, also known as the “Lazarus response”, allowed clinical practice to change. Thus, metastatic NSCLC patients with activating oncogene mutations could receive target therapy irrespective of their PS status [57,58].

Data on the efficacy of ICIs in NSCLC patients with a poor PS are limited and usually come from heterogeneous meta-analyses and small phase II or expanded access trials. Regarding prospective trials, some studies have included PS 2 patients, such as the PePS2 trial with pembrolizumab, the CheckMate 171 and CheckMate 153 trials with nivolumab, and the CheckMate 817 trial with nivolumab plus ipilimumab [59,60,61,62] (Table 1). The PePS2 trial included only patients with PS 2. In the other trials, PS 2 patients were enrolled together with elderly patients and PS ≤ 1 patients affected by comorbidities; therefore, the final results were less specific with regard to the PS 2 patients. However, these trials found grade 3–4 treatment-related adverse events in the PS 2 patients that were similar to those observed in the PS ≤ 1 patients. Conversely, the OS was worse in the PS 2 patients compared to that of the overall population.

A meta-analysis summarized the outcomes of the retrospective studies of first-line immunotherapy in NSCLC patients with a poor PS (PS ≥ 2). In the selected studies, the patients with a poor PS showed worse outcomes in comparison to those with a good PS. In the group with a poor PS, the ORR was 30.9% (vs. 55.2% in the PS ≤ 1 patients), and the disease control rate (DCR) was 41.5% (vs. 71.5% in the PS ≤ 1 patients). Similarly, both the PFS and OS were worse in the patients with a poor PS in comparison to those with a good PS [63].

The “Lazarus response” was also anecdotally described in patients with a very poor PS (PS ≥ 3), who were treated with immunotherapy [64,65]. An initial PS ≥ 3 improved to 0 after only one month of ICI-based therapy, with a follow-up after more than 24 months and a major tumor response. On the basis of these observations, pembrolizumab could be considered to be a valid option for critically ill patients with advanced NSCLC and a PD-L1 expression of ≥50%.

As PS 2 patients represent a heterogeneous population, the main challenge involves knowing how to identify the patients who can really benefit from ICI treatment. This aim could be reached through the identification of ad hoc predictive biomarkers, such as PD-L1, TMB [66], the lung immune prognostic index, inflammatory markers, LDH, and steroids exposure, in addition to those already utilized in clinical practice.

Recently, clinical trials for PS 2 NSCLC patients have been launched and are currently ongoing (ClinicalTrial ID: NCT03620669; ClinicalTrial ID: NCT04108026).

**Table 1 ijms-24-12622-t001:** Selected clinical studies investigating safety and efficacy of ICI-based regimens for advanced NSCLC patients with ECOG performance status > 2 ECOG performance status.

Study	N Patients (PS > 2)	Setting	Drug(s)	ORR (PS > 2)	mPFS (PS > 2)	Ref.
CheckMate171	811 (103)	Pre-treated	Nivolumab	11% (2.6%)	NA (NA)	[59]
CheckMate153	127 (7)	Pre-treated	Nivolumab	68% (NA)	24.7 (34.7)	[67]
CheckMate817	589 (139)	I line	Nivolumab + Ipilimumab	37.3% (20.9%)	5.8 (3.6)	[68]
CheckMate169	161 (30)	Pre-treated	Nivolumab	NA (NA)	NA (NA)	[69]
Immitigata	54 (15)	Pre-treated	Nivolumab	16% (0)	2.5 (1.4)	[70]
Clinivo	902 (121)	Any line	Nivolumab	19% (12.4%)	2.0 (1.7)	[71]
TAIL	615 (61)	Pre-treated	Atezolizumab	11.1% (3.1%)	2.7 (1.7)	[72]
EVIDENS	1420 (192)	Pre-treated	Nivolumab	19.6% (NA)	2.8 (NA)	[73]
PePS2	67 (62)	Any line	Pembrolizumab	NA (NA)	NA (NA)	[60]
SAKK 19/17	21 (21)	I line	Durvalumab	NA (NA)	NA (NA)	[74]
Tabah, J. Clin. Oncol., 2020	254 (34)	I line	Pembrolizumab + CT	NA (NA)	NA (NA)	[75]

## 5. IO in NSCLC with Brain Metastases

The occurrence of brain metastases (BM) is relatively frequent in selected solid tumors, such as lung cancer, breast cancer, and melanoma. It is estimated that up to 40% of lung cancer patients will experience metastatic spreading to the central nervous system (CNS) in the course of their disease [76]. Unfortunately, the prognosis and survival of patients with BM remains poor; the presence of extracranial metastases or leptomeningeal disease, primary disease control, age, and performance status represent the most relevant prognostic factors [77]. No treatments have demonstrated real efficacy for non-addicted lung cancer patients with BM in the pre-immunotherapy era; consistently, their therapeutic options are largely palliative and include surgical resection, whole-brain radiation therapy (WBRT), or stereotactic radiosurgery (SRS) [78], though both WBRT and, to a lower degree, SRS have certain limitations, such as radiation neurotoxicity and cognitive deterioration [79,80,81]. In contrast to radiotherapy, chemotherapy is rarely utilized due to its well-known limitation in effectively crossing the blood–brain barrier (BBB), except for in the case of very limited drugs [82]. Due to the lack of effective treatment, together with the poorer prognosis, in the last decade, the BM patient population has usually been excluded from clinical trials with chemotherapeutic agents, as well as those with immune checkpoint inhibitors (ICIs) [83]. More recently, growing scientific evidence has identified the CNS as immunologically distinct rather than an immune-isolated compartment [84]. The inflammatory TME of BM has been shown to be active in the majority of patients with a dense infiltration of tumor-infiltrating lymphocytes (TILs), which often express immunosuppressive factors such as PD-L1 [85]. This evidence and the availability of effective immunotherapeutic strategies [86,87] targeting CTLA-4, PD-1, and PD-L1 have prompted their use in patients with BM [88], particularly those with negative driver genes [89]. In this regard, Cohen J.V. et al. suggested that ICIs and active T cells could penetrate the BBB [90], which is necessary for ICIs to work.

Limited data are currently available on the clinical efficacy of ICIs in NSCLC patients with BM. The data generated in this scenario are mostly retrospective; they are real-world data and have been preferentially generated in pretreated BM.

For example, clinicopathological features have been retrospectively related to efficacy outcomes in advanced NSCLC patients who receive IO in combination with antiangiogenic drugs. The authors found via a multivariate analysis that brain metastasis represented an independent predictive factor of PFS [91].

In a prospective phase II trial, pembrolizumab induced an intracranial (ic) ORR in 10 out of 34 (29.4%) PD-L1-positive patients, while no objective responses were observed in the 5 PD-L1-negative patient subsets [92]. The median OS among all the patients was 8.9 months, and 31% of the patients were alive after 2 years [92].

A pooled analysis from the three CheckMate (CM) studies (the phase II CM-063 and phase III CM-017 and CM-057) explored the role of nivolumab in second-line NSCLC patients with pretreated BM. The results showed an improvement in survival in patients treated with nivolumab (8.4 months) as compared to those treated with chemotherapy (docetaxel) (6.2 months). Supporting the efficacy of ICIs in this patient population, in 409 NSCLC patients with BM treated with nivolumab within the Italian expanded access program (EAP), the ORR and DCR were 17% and 40%, respectively [93]. Additionally, in the OAK study, an exploratory analysis performed on a cohort of NSCLC patients with no active BM [94] showed an improvement in survival with atezolizumab compared to docetaxel (16 months versus 11.9 months, respectively). Moreover, atezolizumab led to a prolonged time to the radiologic identification of new symptomatic BM compared to the docetaxel arm [95]. A pooled analysis of KEYNOTE-001, KEYNOTE-010, KEYNOTE-024, and KEYNOTE-042 explored the effects of baseline stable BM in both patients with previously treated or untreated PD-L1-positive advanced or metastatic NSCLC who received pembrolizumab monotherapy versus chemotherapy. All the efficacy (PFS and OS) and activity (ORR and duration of response) outcomes were improved by pembrolizumab over chemotherapy, regardless of brain metastasis status [96].

A meta-analysis summarized the outcomes of the three first-line studies (Keynote (KN)-021, KN-189, and KN-407) in the NSCLC patient subpopulation with stable BM compared to those receiving only chemotherapy. Interestingly, the patients who received pembrolizumab alone or combined with chemotherapy received a benefit in terms of their mOS (18.8 months vs. 7.6 months, HR 0.48), mPFS (6.9 months vs. 4.1 months, HR 0.44), and ORR (39% vs. 17.7%), regardless of their PD-L1 status [97]. Similarly, the results generated in the CM-9LA study showed an improvement in mOS (19.3 vs. 6.8 months), mPFS (10.6 vs. 4.1 months), and ORR (43% vs. 24%) in patients with pretreated BM who had received a platinum-based regimen combined with nivolumab plus ipilimumab, compared to those who received only chemotherapy [98]. These results were comparable to the results generated in the IMPower-Lung1 study, in which NSCLC patients with no active BM receiving cemiplimab had a better mPFS compared to those treated with chemotherapy (18.7 months vs. 7.4 months) [46] (Table 2).

Despite these intriguing findings, the efficacy of immunotherapy in BM currently remains controversial due to the limited sample sizes and long-term efficacy data in the above clinical trials, and to the use of various immunotherapy regimens for which there has been no comparison of effectiveness.

In order to overcome these limitations, Chu and colleagues [102] performed a comprehensive meta-analysis that included a total of 3160 participants from 46 trials. The results showed an improvement in PFS (HR = 0.48) and OS (HR = 0.64) for the immunotherapy-based regimen compared to non-immunotherapy in NSCLC patients with BM; this was probably due to the well-known synergy between ICIs and chemotherapy and/or radiotherapy.

Additionally, no significant differences in PFS (HR = 0.97, 95% CI: 0.40–2.35); OS (HR = 0.69, 95% CI: 0.23–1.15); extracranial overall response rate (odds ratios (OR) = 0.75, 95% CI: 0.28–2.01); intracerebral overall response rate (OR = 1.27, 95% CI: 0.65–2.47); intracerebral disease control rate (OR = 1.52, 95% CI: 0.80–2.91); or extracranial disease control rate (OR = 0.99, 95% CI: 0.26–3.81) were observed between ICIs combined with RT and ICI monotherapy. In this regard, future studies should be addressed toward the investigation of both the sequencing of IO and RT and the optimal interval between ICIs and cranial RT in the treatment of BM from NSCLC, in view of their potential influence on the efficacy of ICIs combined with RT. Indeed, the evidence supports concurrent ICIs combined with RT rather than sequential ICIs combined with RT, in terms of a decreased incidence of recurrence. Furthermore, an interval shorter than 2 weeks between ICIs and RT has been associated with a longer OS and PFS. Finally, dual ICIs combined with CT or ICIs combined with CT have provided a better PFS and OS than ICIs alone.

An intriguing, relevant aspect concerns the efficacy of ICIs in NSCLC patients with active BM, for whom limited data are currently available. The data generated in this scenario are mostly retrospective and based on real-world data. However, the evidence shown in metastatic melanoma patients with active BM strongly support the efficacy of ICI treatment, especially in a combination regimen; this has become the new standard of care for metastatic melanoma patients with BM [103,104]. In this regard, and seeking to expand the potential efficacy of ICI treatment in NSCLC patients with active BM, a variety of prospective clinical trials are currently ongoing (Table 3).

An additional major concern is the appropriate evaluation of the radiologic response of brain metastases during treatment with immunotherapy. Indeed, immunotherapy may significantly affect the imaging features of BM, as well as the brain parenchyma, hindering the correct neuroradiological interpretation of post-treatment findings. Consistently, atypical responses, such as initial disease progression or the appearance of new lesions followed by a clinical response and pseudo-progression, can be observed in the course of immunotherapy and misinterpreted as tumor recurrence or progression [105]. This aspect is also more relevant for patients receiving radiotherapy for BM and immunotherapy. Therefore, it is crucial for neuroradiologists to be more comprehensively familiar with the treatment response criteria and treatment-induced changes in brain lesions [106,107,108]. With the aim of standardizing the radiologic evaluation of BM, novel criteria have been proposed and incorporated into the immunotherapy RANO (iRANO) criteria [109,110], providing recommendations for the interpretation of neuroradiological changes in the course of this therapeutic approach [109]. Moreover, PET-based imaging, especially with amino acid tracers, provides information on tumor metabolism and is currently under investigation with regard to the proper differentiation of neoplastic tissues from non-specific, treatment-related changes [111,112,113,114].

In conclusion, the results of ICI-based treatment in NSCLC patients with BM may soon lead to significant changes in their comprehensive management; thus, the roles of surgery and radiotherapy in treating BM may be revisited. Indeed, in selected cases, the ICI-based regimen alone could represent the first, optimal therapeutic choice, though its use requires a careful patient evaluation due to the lack of well-defined selection criteria. Prospective clinical data to corroborate the efficacy of ICI treatment in NSCLC patients with BM are awaited; in the daily practice scenario, a multidisciplinary interaction is mandatory for the optimal management of lung cancer patients with BM and must undoubtedly include a neuroradiologist to support the treating physicians in evaluating clinical responses and neurological side effects.

## 6. Biomarkers

The use of biomarkers could represent an opportunity to stratify patients according to their prognosis or IO efficacy. In this review, we grouped immune-related biomarkers into three categories: biomarkers of tumor immunogenicity (i.e., PD-L1, TMB, and microsatellite instability), biomarkers of the tumor immune microenvironment (i.e., tumor-infiltrating lymphocytes), and biomarkers of the host immune system (i.e., peripheral blood inflammatory markers and myeloid-derived suppressor cells).

### 6.1. Biomarkers of Tumor Immunogenicity

Currently, PD-L1 expression is the most studied predictive biomarker of ICIs targeting the PD-1/PD-L1 axis [115]. Metastatic NSCLC patients achieve a better OS benefit from pembrolizumab in comparison to chemotherapy when their tumors express a PD-L1 of ≥50% [1]. Some data have also highlighted that the higher the PD-L1 expression level, the greater the benefit from anti-PD-1/PD-L1, particularly when used as a monotherapy [41]. Despite these findings, PD-L1 expression has great limitations as a biomarker, given that benefits from immunotherapy are also observed in patients bearing PD-L1-negative tumors. For this reason, research on further biomarkers is ongoing.

TMB represents the total amount of DNA mutation per megabase. Only nonsynonymous tumor mutations are considered. This amount is supposed to be related to the generation of neoantigens that T lymphocytes can recognize as non-self. A higher number of neoantigens could enhance the efficacy of ICI-based therapy. TMB is independent of PD-L1 expression [116]. However, a high TMB is more related to the efficacy of the combination of nivolumab plus ipilimumab in NSCLC patients with negative PD-L1 [117]. Moreover, a combination of the CTLA-4 inhibitor tremelimumab plus the PD-L1 inhibitor durvalumab did not improve OS in comparison to chemotherapy, but a better OS was observed in patients with a high TMB in their ctDNA; these patients were treated with this combination immunotherapy [118]. In the B-F1RST phase 2 trial, blood-based TMB was evaluated as a predictive biomarker for first-line monotherapy with atezolizumab in advanced NSCLC patients. A TMB of ≥16 was associated with higher tumor responses, which further increased at higher TMB cutoffs. The OS was also better with a TMB of ≥16 in an exploratory analysis of this trial [119]. In a meta-analysis by Ma et al., including almost three thousand patient tumor responses, the PFS and OS were improved in the group of patients with a high TMB in comparison to those with a low TMB [120]. However, the use of TMB as a biomarker is still limited by its heterogeneity across the various NSCLC subtypes and variable detection assays, and the lack of a cutoff standardization.

Microsatellite instability (MSI) is the genomic consequence of mismatch repair deficiency (dMMR). Microsatellites are short tandem repeats present throughout the genome. The instantaneous dissociation of the DNA strand during replication can change the microsatellite lengths, which should be corrected by the mismatch repair system. MSI severity can be categorized into three groups: microsatellite stable (MSS), MSI low (MSI-L), and MSI high (MSI-H) [121,122]. MSI and dMMR involve a high tumor immunogenicity; thus, some authors have hypothesized that they have a role as predictive biomarkers for ICI-related outcomes. The combined results of the study on ICI efficacy emerged in MSI-H tumors, irrespective of tumor type [123]. These findings led the FDA to approve pembrolizumab for the treatment of patients with advanced solid tumors when an MSI-H or dMMR status was present. Prospective randomized studies focused on NSCLC are needed in order to better identify further applications for these biomarkers.

Many efforts have also been made to standardize the analysis of these biomarkers in the peripheral blood instead of tumor tissue. These include TMB in cell-free DNA from the plasma of NSCLC patients treated with atezolizumab as subsequent-line treatment [124], or with pembrolizumab-based first-line treatment [125], soluble PD-1/PD-L1 [126], PD-L1 mRNA and exoPD-L1 [127], or PD-L1-positive circulating tumor cells [128]. Even though some of these biomarkers have achieved a prognostic impact in patients treated with IO, none of these have reached an immediate applicability in clinical practice.

### 6.2. Biomarkers of Tumor Immune Microenvironment

The presence of CD8^+^ cytotoxic T lymphocytes infiltrating the tumor stroma (TILs) is a requirement for ICI anti-tumor action. Some studies have already shown the favorable prognostic role of TILs in NSCLCs [129,130]. These findings led to the hypothesis that these cells could also represent a predictive biomarker of ICI efficacy [131]. Various approaches have been used to address this aim. Among these, RNA sequencing and immunohistochemistry staining of NSCLC samples from patients treated with anti-PD-1 showed that high CD8A and CD274 mRNA expressions were associated with a longer PFS [132]. TILs were also taken into account as an Immunoscore to supplement TNM staging in NSCLC, called TNM-Immune (TNM-I) [133].

In the KEYNOTE-001 phase I trial, the tumor responses to pembrolizumab were associated with a higher quantity of TILs in baseline tumor biopsies. For this purpose, the biopsy slices were stained for CD8 [134]. Similarly, the search for TILs in tissue samples from metastatic NSCLC or melanoma patients treated with anti-PD1 drugs highlighted that response rates increased with CD8^+^ lymphocyte count and CD8^+^/CD4^+^ ratios [135]. Other authors have used multiplexed quantitative immunofluorescence to characterize both PD-L1 expression and TILs and their state of activation; they have also been characterized in relation to their mutational status. NSCLC tissues bearing a KRAS mutation were more inflamed because of a greater quantity of active TILs. However, EGFR mutant tumors hosted inactive TILs. Moreover, activated TILs were related to a higher PD-L1 expression, only in tumors without EGFR or KRAS mutations [136]. Some authors have also used the gene expression signature of CD8^+^ T lymphocytes in correlation with a radiomic signature for the detection of CD8^+^ TILs. The aim of this association was an indirect estimation of the presence of TILs through a computed tomography scan to predict responses to ICI-based therapy [137]. A new technique is under development using CD8 PET imaging with the ^89^Zr-Df-IAB22M2C radioisotope to visualize the distribution of CD8^+^ T cells in the whole body or at tumor sites, and potentially to predict early tumor responses to immunotherapy [138].

### 6.3. Biomarkers of Host Immune System

Currently, a lot of evidence is available regarding systemic inflammation markers in the peripheral blood as prognostic or predictive factors, particularly in metastatic NSCLC patients treated with ICI-based therapy. The neutrophil-to-lymphocyte ratio (NLR) and platelet-to-lymphocyte ratio (PLR) are the most studied. These biomarkers are based on the association between tumor development and increased inflammation and can be easily extracted from routine blood tests. Thus, these markers are usually available worldwide and are highly reproducible with no further costs [139,140]. Many studies on this topic have been carried out; then, these studies have been summarized in some meta-analyses, which confirmed that NLR and PLR may be considered as prognostic factors in metastatic NSCLC patients treated with ICI-based therapy [141,142]. The former, by Tan et al. [141], suggested that a high baseline NLR predicted a worse PFS and OS, but this result was not confirmed for PLR. The latter, by Platini et al. [142], found that both biomarkers were prognostic in the same setting.

Peripheral blood inflammatory markers were combined with other parameters to obtain prognostic scores in order to predict the outcomes in metastatic NSCLC patients treated with ICIs. These scores included further laboratory variables, e.g., the derived NLR (dNLR), calculated with the formula [neutrophils/leucocytes–neutrophils], lactate dehydrogenase (LDH), albumin, and C-reactive protein, as well as clinical variables, e.g., ECOG PS or tumor stage. Twenty-two combined scores were studied on the basis of baseline values used only for prognosis. Some of these also showed a predictive value for ICI-based therapy in both pretreated patients and the first-line setting or treatment monitoring [143].

Among these prognostic scores, the lung immune prognostic index (LIPI) has been the most studied in metastatic NSCLC patients. It includes two parameters, the dNLR (cutoff: three) and LDH (cutoff: above the limit of normal). The presence or absence of each factor defines a score for prognostic stratification, which is categorized into three groups: a good LIPI for both factors under the cutoff; an intermediate LIPI for one factor above the cutoff; and a poor LIPI for both factors above the cutoff [144]. Some subgroup analyses of randomized trials and a pooled analysis including the above prognostic score have confirmed its prognostic role in patients treated with both ICI and chemotherapy, but not its predictive usefulness [145,146].

The presence or activation of suppressive cells can also limit the efficacy of ICIs. Myeloid-derived suppressor cells (MDSCs) are the most studied and have shown a prognostic role both in treatments with chemotherapy and in those with ICIs. MDSCs are immature myeloid cells, which usually increase with tumor progression and are related to systemic chronic inflammation. MDSCs interact with the host immune system through various mechanisms: the inhibition of T cell function via the depletion of some fundamental amino acids in the proliferation of T lymphocytes; the interference of the PD-1/PD-L1 signaling pathway with T cell viability and relative migration; the production of nitric oxide (NO) and reactive oxygen species (ROS), which induces T cell apoptosis; the transition of CD4^+^ T cells into regulatory T cells (Tregs) via TGF-β; the repolarization of macrophages towards an M2 phenotype; the impairment of natural killer (NK) cell function via direct cell–cell contact; and the reduction in IFN-γ production [147,148,149,150].

MDSCs can be grouped into two subpopulations: polymorphonuclear cells (PMN-MDSCs), characterized by CD11b^+^CD14^−^CD15^+^ or CD11b^+^CD14-CD66b^+^, and monocytic cells (M-MDSC), characterized as CD11b^+^CD14^+^HLA-DR^−/low^CD15 cells. These surface markers have great limitation in that they cannot help to distinguish normal monocytes from M-MDSCs and neutrophils from PMN-MDSCs. A unique pattern of markers specific for MDSCs is not currently available [151]. However, these cells have the advantage that they can be studied in the peripheral blood as circulating markers.

A recent meta-analysis highlighted that NSCLC patients with high circulating M-MDSC levels achieved a statistically significant shorter PFS and OS than patients with low levels, irrespective of treatment. Statistical significance was not reached for PMN-MDSCs [152]. Among the 14 studies summarized in this meta-analysis, 3 considered NSCLC patients treated with immunotherapy and included 1 study on PMN-MDSCs, 1 on M-MDSCs, and 1 on both cell subpopulations. The first one achieved a better OS when the circulating PMN-MDSCs were higher [153]. In the second one, the OS was worse, with high circulating M-MDSCs [154]. In the latter, a worse OS was associated with high PMN- and M-MDSCs [155]. Given that these cells could also be targeted through various strategies, the research on MDSCs treated with ICI-based therapy is still ongoing [156].

## 7. Conclusions

In the last few years, we have witnessed an impressive therapeutic paradigm shift in NSCLC with the appearance of the IO-based regimen in all clinical settings. However, the role of these IO regimens has not been fully established for a significant proportion of patients, particularly elderly patients with cases complicated by BM and/or TKI-resistant driver gene mutations; it remains a challenge to treat these patients. The knowledge on the applicability of immunotherapy in “special” NSCLC populations is mostly derived from studies with other purposes. The various studies discussed in this paper highlight how ICI-based therapy is not precluded in these patients.

A broader understanding of immune and inflammatory responses will fully allow a definition of the real benefit of IO-based treatment for “special” NSCLC populations, as will the ad hoc design of combined/sequential therapies. The availability of predictive/prognostic biomarkers could help to select the patients for whom immunotherapy would actually be beneficial, rather than those patients belonging to a “special population”. Currently, we have some data on the biomarkers of tumor immunogenicity (i.e., TMB and MSI), the tumor immune microenvironment (i.e., TILs), and the host immune system (i.e., NLR, PLR, LIPI, and MDSCs). However, these biomarkers have not yet been sufficient to stratify patients belonging to ‘special populations’. Additionally, the identification of validated biomarkers via multi-omics approaches will also be mandatory for the better selection of NSCLC patients for IO therapy.

## Figures and Tables

**Table 2 ijms-24-12622-t002:** Selected clinical studies investigating efficacy and safety of ICI-based regimens in NSCLC patients with pre-treated, stable brain metastases.

Study	N Patients (BMs)	Setting	Drug(s)	ORR (BMs)	mPFS (BMs)	mOS (BMs)	%G3–4 irAEs (BMs)	Ref.
Goldberg, 2018	42 (42)	Any line	Pembrolizumab	29.4 (ic)	1.9 (1.9)	8.9 (8.9)	21 (21)	[92]
Italian EAP Nivolumab	1588 (409)	Any line	Nivolumab	NA (17)	NA (NA)	NA (8.1)	NA (NA)	[99]
OAK Exploratory Study	425 (61)	Pre-treated	Atezolizumab	NA (NA)	NA (NA)	13.2 (16.0)	15 (15)	[95]
Keynote 021–189-407	1298 (171)	I line	Pembrolizumab + CT	54.6 (39)	8.8 (6.9)	22.5 (18.8)	50.5 (59.8)	[97]
CheckMate 9LA-BMs	361 (51)	I line	Nivolumab + Ipilimumab + CT	37 (43)	5.8 (10.6)	15.6 (19.3)	NA (NA)	[100]
EMpower-Lung 1 BMs	563 (68)	I line	Cemiplimab	NA (41.2)	8.2 (10.4)	23.4 (18.7)	29.7 (NA)	[101]

**Table 3 ijms-24-12622-t003:** Selected, ongoing clinical trials investigating efficacy and safety of ICI-based regimens in NSCLC or melanoma patients with untreated brain metastases.

Study Name	Status	Primary Tumor	Treatment	Study ID (NCT)	N Patients	Primary Endpoint
USZ-STRIKE	Recruiting	NSCLC, Melanoma	ICIs ± SRS	NCT05522660	190	CNS-PFS
Durvalumab and Radiosurgery for NSCLC BMs	Recruiting	NSCLC	Durvalumab + SRS/ Durvalumab + Pulsar	NCT04889066	46	Intracranial clinical benefit
Pembro + Chemo in Brain Mets	Recruiting	NSCLC	Pembrolizumab + CT	NCT04964960	45	DCR
Beva + Atezo **±** Cobimetinib in Brain Mets	Recruiting	Melanoma	Bevacizumab+ Atezolizumab ± RT	NCT03175432	60	icORR

## Data Availability

Not applicable.

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
