# Peer review of "Immune Checkpoint Inhibitors in “Special” NSCLC Populations: A Viable Approach?"

_ijms, 2023, doi:10.3390/ijms241612622_

Round 1

Reviewer 1 Report

This review highlights the need to elucidate the real benefit of immunotherapy (IO) in selected NSCLC patient cohorts (elderly, brain metastases or oncogene-addicted mutations). Additionally, the review also provides a brief summary of ‘predictive biomarkers’ of NSCLC. Whilst comprehensive, the review needs major changes to improve cohesiveness and flow before being considered for acceptance.  

·   1. The introduction needs to be revised to improve clarity. The authors try to illustrate that whilst IO has been shown to benefit/improve OS of NSCLC patients, the efficacy of IO in specific NSCLC patient cohorts (elderly, never-smoked, oncogene-addicted, etc. patients) needs further elucidation/validation. However, this point is not immediately clear – to improve clarity the authors could consider rearranging the information in more cohesive manner, define what they mean by most common ‘special’ NSCLC subpopulations, and indicate the purpose of this review.

·    2. The authors may have misinterpreted the original study in lines 53-55. This reference which is a poster abstract seems to indicate that patients aged >75 years has similar responses and not “…in patients with PD-L1 ≥ 50% has shown that first-line IO alone or in combination with chemotherapy yields a similar OS”. Please check.

·  3. In section 2 (IO in never smoker and/or oncogene-addicted NSCLC patients) – the discussion of TKI seems the focus of this section seems to be more on discussing TKI than on IO, it would be better if the authors rearrange this section to introduce oncogene-addicted NSCLC, clinical efficacy of IO reported in this subgroup, TKI, IO combinations with TKI etc and limitations/future studies needed.

·    4. In section 3 (IO in elderly NSCLC patients) please include some discussion about IO efficacy in younger and older adults with advanced NSCLC. Please discuss the findings of Reck et al [27], Nosaki et al [89], Akinboro et al [8], Herbst et al [13], Sezer et al [74]. Please include the following references in this section: PMID: 37076834, 34734989, 30113497, 33072584, 30955977, 30476576, and doi.org/10.1016/j.annonc.2022.08.052.

·    5. Please add and discuss PMID: 37208639 and 34590048 in section 5 (IO in NSCLC with brain metastases)

·   6. The shift from IO in different NSCLC settings to biomarkers of IO is too abrupt. Authors could consider moving the biomarkers subsections under a major subheading ‘Biomarkers’ with a short introduction to improve the flow.

·   7. It would be better and more prudent to retain the focus of this review on ‘special NSCLC subsets’ by discussing biomarkers in the context of helping better stratification/prognosis of these patients in addition to providing on overview of biomarkers being currently utilized/proposed.

·  8. In ‘Biomarkers of tumor immune microenvironment’ section please include PMID: 29388007 as well as discuss use of ‘Immunoscore’ prognostic tool in NSCLC [PMID: 26578726].

·  9. Please include use of circulating tumor cells (CTCs), cfDNA etc for assessing PD-L1 expression, TMB and other immune-related signatures in the biomarker section [PMID: 30082870, 29228642, 2627341, 32102950, 33936103, and 33918147].

·    10. There are some statements throughout the review describing other work that are missing the relevant references (e.g.: lines 48-51) and minor inconsistencies with in-text reference format used (line 332)

·    11. Please correct typo in line 46: it seems clear that IO is not fit for all comers

Use of shorter sentences and minor English editing could be considered. Please correct typo in line 46: it seems clear that IO is not fit for all comers. 

Author Response

Reviewer 1

Comments and Suggestions for Authors

This review highlights the need to elucidate the real benefit of immunotherapy (IO) in selected NSCLC patient cohorts (elderly, brain metastases or oncogene-addicted mutations). Additionally, the review also provides a brief summary of ‘predictive biomarkers’ of NSCLC. Whilst comprehensive, the review needs major changes to improve cohesiveness and flow before being considered for acceptance. 

  • 1. The introduction needs to be revised to improve clarity. The authors try to illustrate that whilst IO has been shown to benefit/improve OS of NSCLC patients, the efficacy of IO in specific NSCLC patient cohorts (elderly, never-smoked, oncogene-addicted, etc. patients) needs further elucidation/validation. However, this point is not immediately clear – to improve clarity the authors could consider rearranging the information in more cohesive manner, define what they mean by most common ‘special’ NSCLC subpopulations, and indicate the purpose of this review.

REPLY: We appreciate this valuable comment of the reviewer. We modified the last paragraph of the Introduction section according to the suggestions.

  • 2. The authors may have misinterpreted the original study in lines 53-55. This reference which is a poster abstract seems to indicate that patients aged >75 years has similar responses and not “…in patients with PD-L1 ≥ 50% has shown that first-line IO alone or in combination with chemotherapy yields a similar OS”. Please check.

REPLY: the reference 8 was checked and corrected. The comment on the results of this pooled analysis was rephrased according to a more correct interpretation.

  • 3. In section 2 (IO in never smoker and/or oncogene-addicted NSCLC patients) – the discussion of TKI seems the focus of this section seems to be more on discussing TKI than on IO, it would be better if the authors rearrange this section to introduce oncogene-addicted NSCLC, clinical efficacy of IO reported in this subgroup, TKI, IO combinations with TKI etc and limitations/future studies needed.

REPLY: We deleted some phrases in this section for a better focus on IO in this subgroup rather than discussing TKI. We also included a sentence at the end of this section to make it clearer that immunotherapy has a minor role in oncogene-addicted patients on the basis of retrospective and phase II studies and subgroup analyses of some trials.

  • 4. In section 3 (IO in elderly NSCLC patients) please include some discussion about IO efficacy in younger and older adults with advanced NSCLC. Please discuss the findings of Reck et al [27], Nosaki et al [89], Akinboro et al [8], Herbst et al [13], Sezer et al [74]. Please include the following references in this section: PMID: 37076834, 34734989, 30113497, 33072584, 30955977, 30476576, and doi.org/10.1016/j.annonc.2022.08.052.

REPLY: The results of Reck’s trial were updated with those more recent in Socinski et al. 2021 [PMID: 34311108]. The results of Herbst’s trial and Mok’s trial [PMID: 30955977] about age subgroups are included in the pooled analysis by Nosaki, which was also discussed in section 3. Both the papers by Akinboro and Sezer were discussed in the same section among the studies in the front-line setting. The other additional suggested references were added and discussed in this section.

  • 5. Please add and discuss PMID: 37208639 and 34590048 in section 5 (IO in NSCLC with brain metastases)

REPLY: The suggested references were added and discussed in section 5.

  • 6. The shift from IO in different NSCLC settings to biomarkers of IO is too abrupt. Authors could consider moving the biomarkers subsections under a major subheading ‘Biomarkers’ with a short introduction to improve the flow.

REPLY: We added a brief introduction to the section 6 (biomarkers) and changed the subsequent subheadings.

  • 7. It would be better and more prudent to retain the focus of this review on ‘special NSCLC subsets’ by discussing biomarkers in the context of helping better stratification/prognosis of these patients in addition to providing on overview of biomarkers being currently utilized/proposed.

REPLY: A sentence was added in section 6 to indicate that ‘The use of biomarkers could represent an opportunity to stratify patients according to their prognosis or IO efficacy.’

  • 8. In ‘Biomarkers of tumor immune microenvironment’ section please include PMID: 29388007 as well as discuss use of ‘Immunoscore’ prognostic tool in NSCLC [PMID: 26578726].

REPLY: The suggested references were added and discussed in section 6.2.

  • 9. Please include use of circulating tumor cells (CTCs), cfDNA etc for assessing PD-L1 expression, TMB and other immune-related signatures in the biomarker section [PMID: 30082870, 29228642, 2627341, 32102950, 33936103, and 33918147].

REPLY: A paragraph was added at the end of section 6.1 briefly describing the use of biomarkers of tumor immunogenicity from peripheral blood, and the suggested references were cited, except for reference PMID: 2627341, which is not related to this topic (maybe it was suggested for a mistake).

  • 10. There are some statements throughout the review describing other work that are missing the relevant references (e.g.: lines 48-51) and minor inconsistencies with in-text reference format used (line 332)

REPLY: Reference in the first paragraph of section 2 was added. The inconsistency of in-text reference format was corrected.

  • 11. Please correct typo in line 46: it seems clear that IO is not fit for all comers

REPLY: This sentence was rephrased.

Comments on the Quality of English Language:

Use of shorter sentences and minor English editing could be considered. Please correct typo in line 46: it seems clear that IO is not fit for all comers.

REPLY: The manuscript has undergone English language editing by MDPI. The text has been checked for correct use of grammar and common technical terms, and edited to a level suitable for reporting research in a scholarly journal.

Reviewer 2 Report

Content of the article by Bronte and colleagues is appropriate and presented in a well written format, with only some minor expansion and clarifications requested, as indicated below.

-(Lines 121-128) Please revise this passage to ensure it is very clear what is being described. As written, there are multiple interpretations on treatments and what is being compared.

-(Section 3) Intricacies for the differential outcomes with elderly patients are being increasingly appreciated in recent years. The authors provide a nice review of cases receiving IO...though it would be nice to have a better appreciation of outcomes in similar populations in regards to other treatments (EGFR-TKIs, ALK-inh., etc.) for comparison. 

- In all tables listing clinical trials, please provide the reference number for the bibliography to aid the audience.

- description of biomarkers offered little information over other reviews. Suggest condensing this section a tad and/or add information relevant to the special populations defined in the other sections.

Author Response

Reviewer 2

Comments and Suggestions for Authors

Content of the article by Bronte and colleagues is appropriate and presented in a well written format, with only some minor expansion and clarifications requested, as indicated below.

-(Lines 121-128) Please revise this passage to ensure it is very clear what is being described. As written, there are multiple interpretations on treatments and what is being compared.

REPLY: In this paragraph the kind of treatments, which obtained those results, was specified.

-(Section 3) Intricacies for the differential outcomes with elderly patients are being increasingly appreciated in recent years. The authors provide a nice review of cases receiving IO...though it would be nice to have a better appreciation of outcomes in similar populations in regards to other treatments (EGFR-TKIs, ALK-inh., etc.) for comparison.

REPLY: We added a paragraph in section 3 to explain the effect of TKIs in elderly population.

- In all tables listing clinical trials, please provide the reference number for the bibliography to aid the audience.

REPLY: Reference number were added in all tables.

- description of biomarkers offered little information over other reviews. Suggest condensing this section a tad and/or add information relevant to the special populations defined in the other sections.

REPLY: We added a sentence in the conclusions sections regarding biomarkers, because data are not sufficient to stratify the patients in ‘special populations’.
